# Carboxymethylated *Rhizoma alismatis* Polysaccharides Regulate Calcium Oxalate Crystals Growth and Reduce the Regulated Crystals’ Cytotoxicity

**DOI:** 10.3390/biom13071044

**Published:** 2023-06-27

**Authors:** Xiao-Yan Cheng, Jian-Ming Ouyang

**Affiliations:** Institute of Biomineralization and Lithiasis Research, College of Chemistry and Materials Science, Jinan University, Guangzhou 510632, China

**Keywords:** *Rhizoma alismatis* polysaccharide, carboxymethylation, calcium oxalate, cytotoxicity, kidney stones

## Abstract

Objective: This study explored the effects of polysaccharides (RAPD) extracted from the traditional anti-stone Chinese medicine *Rhizoma alismatis* and their carboxymethylated derivatives (RAPs) on the crystal phase, morphology, and size of calcium oxalate (CaOx). It also determined the damaging ability of the regulated crystals on human renal tubular epithelial cells (HK-2). Methods: RAPD carboxymethylation with a carboxyl group (–COOH) content of 3.57% was carried out by the chloroacetic acid solvent method. The effects of –COOH content in RAPs and RAP concentration on the regulation of CaOx crystal growth were studied by controlling the variables. Cell experiments were conducted to explore the differences in the cytotoxicity of RAP-regulated crystals. Results: The –COOH contents of RAPD, RAP1, RAP2, and RAP3 were 3.57%, 7.79%, 10.84%, and 15.33%, respectively. RAPs can inhibit the growth of calcium oxalate monohydrate (COM) and induce the formation of calcium oxalate dihydrate (COD). When the –COOH content in RAPs was high, their ability to induce COD formation was enhanced. In the crystals induced by RAPs, a high COD content can lower the damage to cells. In particular, the cytotoxicity of the crystals induced by RAP3 was the lowest. When the concentration of RAP3 increased, the cytotoxicity gradually increased due to the reduced size of the formed COD crystals. An interaction was observed between RAPs and crystals, and the number of RAPs adsorbed in the crystals was positively correlated with the –COOH content in RAPs. Conclusions: RAPs can reduce the damage of CaOx to HK-2 cells by regulating the crystallization of CaOx crystals and effectively reducing the risk of kidney stone formation. RAPs, especially RAP3 with a high carboxyl group content, has the potential to be developed as a novel green anti-stone drug.

## 1. Introduction

Kidney stones are the product of pathological mineralization [1] and have an incidence rate of up to 14.8%, which increases annually [2]. Calcium oxalate (CaOx) is the most common kidney stone component, accounting for about 70%. Kidney stone formation is a complex physiologically regulated process and mainly includes nucleation, growth, aggregation, and crystal-cell adhesion of urinary microcrystals [3,4]. Urine components and their interactions with urine microcrystals affect these processes. Crystals adhering to the surface of renal tubular epithelial cells can trigger cell inflammation and accelerate stone formation [5].

The existing form of CaOx crystals is an important factor affecting the formation of stones [4]. Calcium oxalate monohydrate (COM) has the highest thermodynamic stability and affinity with renal epithelial cells and cytotoxicity [6]. Calcium oxalate dihydrate (COD) has weaker adhesion ability than COM and is easily excreted in urine [7]. Therefore, the clinical discovery frequency of COM is about twice that of COD [8], and the percentage of COD in the urine of healthy people is higher than that in patients with urolithiasis.

Healthy human urine contains many inhibitors with acidic groups, such as glycosaminoglycans (GAGs), Tamm-Horsfall protein (THP), and citrate [9]. These substances can bind to Ca^2+^ ions in urine and reduce the supersaturation of urine, thereby reducing the growth rate of crystals and inducing the formation of crystals with small particles [10,11]. Moreover, these substances can be adsorbed on the active site on the crystal surface, inhibit the growth of COM crystals, and induce the formation of COD crystals that are easily discharged in vitro [12]. The adsorption of inhibitors on the crystal surface makes the zeta potential negative and enhances the repulsive force between crystals. All these effects are beneficial to reduce the risk of kidney stone formation [13].

Semangoen et al. [14] found that COD could adhere to renal epithelial cells, but that it does not cause cell death. Vinaiphat et al. [15] studied the lithiasis of different types of stones and found that COM is more cytotoxic than COD. Compared with COD, COM induces more cell death and has a higher cell affinity, higher degree of adhesion, and faster crystal growth rate. Therefore, it has a higher risk of lithiasis.

The organic matrix in kidney stones accounts for 2–3% of the total weight and is mainly composed of macromolecules in urine, especially proteins [16]. These substances play an important role in stone formation [17] because they affect the properties and size of the microcrystals deposited in urine, thus influencing whether the crystals can be retained in the kidney. Doyle et al. [18] isolated a protein (CME) from CaOx crystals in human urine. In-vitro tests showed that CME could effectively reduce the growth rate of crystals in urine and inhibit the aggregation of crystals, thus remarkably reducing the possibility of CaOx crystal retention in vivo.

Poon et al. [19] studied the effect of GAG on CaOx crystallization. With their high negative charge, GAGs can combine with Ca^2+^ to reduce urine saturation, thus inhibiting the growth rate of crystals and generating small CaOx crystals. The carboxyl or sulfuric acid groups on the exposed sugar residues in the GAG chain can prevent CaOx crystals from binding to each other, effectively inhibiting inter-crystal aggregation. Duan et al. [20] found that after patients with stones ingested potassium citrate, the supersaturation of urine decreased due to Ca^2+^ chelation in urine, leading to a decrease in the growth rate of CaOx crystals. Ryall et al. [21] studied the effects of pyrophosphate, citrate, and GAGs in urine on the growth rate and aggregation of CaOx crystals in vitro and found that chondroitin sulphate (a GAG) has the most significant inhibitory effect on crystal aggregation; other components could play a synergic role as well.

Natural plant polysaccharides have attracted attention because of their superior biocompatibility and biodegradability. However, their biological activity is often limited by their large molecular weight and low content of functional groups. Chemical modification is an effective method for natural polysaccharides, with the most common techniques including sulfation, carboxymethylation, phosphorylation, and selenization [22,23,24,25,26]. The biological activities of modified polysaccharides (such as antioxidant activity, anticancer activity, and immune activity) are often greatly improved. For example, sulfated *Undaria pinnatifida* polysaccharides [27] and *Porphyra yezoensis* polysaccharides [28] show potential to inhibit kidney stone formation because sulfated polysaccharides contain many polyanions, which could chelate solution Ca^2+^ ions and reduce CaOx saturation in urine. In addition, polyanionic polysaccharides can enhance the surface charge of crystals and reduce crystal mutual aggregation.

*Rhizoma alismatis* has a long history of use in China and exhibits physiological activities, such as diuretic, anti-urolithiasis, antioxidant, and anti-inflammatory characteristics [29,30]. Du et al. [31] showed that *R. alismatis* extract is almost nontoxic to cells at low concentrations, and that adding it to oleic acid-treated hepatocytes could reverse the lipid peroxidation in hepatocytes and reduce the expression of cellular inflammatory factors, as well as the degree of cell damage. Kim et al. [32] found that the ethanol extract of *R. alismatis* could significantly alleviate the pathological features in a pulmonary disease mouse model by reducing the expression of pro-inflammatory cytokines, including TNF-α, IL-6, and TGF-β. Treatment with ethanol extract *R. alismatis* can also downregulate inflammation and reduce lipid metabolism, as well as renal fibrosis in rats with chronic kidney disease [33]. Cao et al. [34] found that ingesting *R. alismatis* decoction can significantly promote the excretion of upper urinary tract stones and reduce the incidence of renal colic in patients with calculi. Yin et al. [35] found that the water extracted from *R. alismatis* can significantly inhibit the growth and aggregation of CaOx crystals in vitro and significantly reduce the content of renal calcium and the formation of CaOx crystals in the renal tubules in vivo, thereby inhibiting the formation of kidney stones in rats.

Although many investigations have focused on *R. alismatis* extract, *R. alismatis* polysaccharide (RAP), one of the important active components of *R. alismatis*, has rarely been studied [30]. Qian et al. [36] found that RAP could increase the glucose and lipid metabolism in diabetic mice and improve the symptoms of mice.

In this work, different degrees of carboxymethylation were performed on natural RAP to enhance its biological activity. The regulatory effect of RAPs on CaOx crystal growth and the difference in toxicity to renal tubular epithelial cells among the formed crystals before and after modification were investigated. This research can further improve the efficacy of the basic components of *R. alismatis* and provide inspiration for the development of new green nontoxic anti-stone drugs.

## 2. Experimental Section

### 2.1. Materials and Apparatus

*R. alismatis* was provided by Shanxi Ciyuan Biotechnology Co., Ltd., (Xian, China); its molecular weight was 46.2 kDa. The glucose, galactose, fucose, rhamnose, arabinose, mannose, fructose, xylose, ribose, glucuronic acid, galacturonic acid, mannuronic acid, guluronic acid standards, and deuterated water (99.9%) were purchased from Sigma Chemical Co., Ltd. (Temecula, CA, USA). Trifluoroacetic acid, isopropanol, chloroacetic acid, FeSO_4_, phenanthroline, hydrogen peroxide, 1,1-diphenyl-2-trinitrophenylhydrazine (DPPH), sodium oxalate and calcium chloride, chloroform, and 1-butanol were purchased from Guangzhou Chemical Reagent Factory (Guangzhou, China); the experimental water is double distilled water.

Human renal proximal tubule epithelial (HK-2) cells were provided by Shanghai Cell Bank, Chinese Academy of Sciences (Shanghai, China). Fetal bovine serum (FBS), DMEM-F12 cell culture medium, penicillin, streptomycin were purchased from Gibco Biochemical Products Co., Ltd. (Carlsbad, CA, USA). Cell proliferation assay kit (Cell Counting Kit-8, CCK-8), reactive oxygen detection kit (DCFH-DA) were purchased from Shanghai Biyuntian Biotechnology Co., Ltd. (Shanghai, China). Twelve- and 96-well cell culture plates, Cell Viability/Cytotoxicity Assay Kit (Calcein-AM/PI) (Keygen) were purchased from Guangzhou Jetway Biotechnology Co., Ltd. (Guangzhou, China).

Fourier-transform infrared spectrometer (FT-IR, Bruker, Ettlingen, Germany), Varian Bruker-500 MHz nuclear magnetic resonance (Wetzlar, Germany), Cary 500 UV-Vis Spectrophotometer (Agilent, Santa Clara, CA, USA), D/max 2400 X-ray powder diffractometer (Rigaku, Tokyo, Japan), ULTRA55 field emission scanning electron microscope (Eindhoven, The Netherlands), OPTIMA-2000DV inductively coupled plasma (ICP, CA, USA), Thermogravimetric analyzer (TGA/DSC 3+, Mettler Toledo, Zurich, Switzerland), Enzyme mark instrument (Safire2, Tecan, Männedorf, Switzerland), Zetasizer Nano ZS laser nanoparticle size analyzer (Malvern, Worcestershire, UK), Inverted fluorescence microscope (Leica DMRA2, Wetzlar, Germany).

### 2.2. Carboxymethylation and Characterization of R. alismatis Polysaccharides

#### 2.2.1. Extraction, Purification and Separation of RAP

The polysaccharide was extracted by hot water extraction method. First, the *R. alismatis* extract (10 g) was added to 95% ethanol (50 mL, 1:5, *w*:*v*) to defat overnight, and the residue was collected by centrifugation. The collected material was then added to ultrapure water (1:12, *w*:*v*) and extracted twice at 90 °C for 2 h each time; the obtained supernatant was concentrated, and crude natural RAP was obtained by precipitation with five volumes of ethanol and centrifugation. The protein was removed by the Sevage method. Crude natural *R. alismatis* polysaccharide was prepared into 5% aqueous solution, and 1/3 volume of Sevage reagent (*V*_chloroform_: *V*_1-butanol_ = 4:1) was added. The solution was shaken fully for 30 min, while the protein was denatured, precipitated, and then centrifuged. Then, the supernatant was transferred to a 3000 Da dialysis bag for 3 d of dialysis and then concentrated, precipitated by adding ethanol, filtered, and dried to obtain natural RAP [37].

#### 2.2.2. Degradation of RAP

RAP (1.2 g) and distilled water (25 mL) were mixed and placed in a water bath at 70 °C and stirred for 30 min until the solid was completely dissolved. The temperature was raised to 90 °C, 4% H_2_O_2_ was rapidly added, and the reaction was continued for 2 h. After cooling to room temperature, the pH of the system was adjusted to 7 with 2 mM of NaOH, then evaporated and concentrated at 50 °C to a volume of 5–10 mL. After cooling, five times the volume of ethanol was added to precipitate, and the degraded polysaccharide (RAPD) was obtained by filtration.

#### 2.2.3. Monosaccharide Composition Analysis of RAPD

Referring to the literature [37], RAPD (5 mg) was hydrolyzed with 2 M of trifluoroacetic acid (TFA, 1 mL) at 121 °C for 2 h. Then dried with nitrogen and methanol was added, washed, and blow dried, repeated three times. Sterile water was added to dissolve, and residue was filtered on Thermo ICS-5000 HPIC system (Thermo Fisher Scientific, San Jose, CA, USA) and equipped with high-performance anion-exchange chromatography (HPAEC) and Dionex™ CarboPac™ PA20 (150 mm × 3.0 mm, 10 μm) liquid chromatography column.

The monosaccharide composition and content were determined by comparing the retention times and peak areas of the samples with those of monosaccharide standards.

#### 2.2.4. Carboxymethylation of RAPD

Dried RAPD (500 mg) and isopropyl alcohol (15 mL) were added to the beakers, then stirred at room temperature for 15 min. The temperature was raised to 60 °C; 30% (*w*/*v*) NaOH solution (10 mL) was slowly added. After the solution was stirred until completely dissolved, a certain amount of chloroacetic acid (dissolved in 2 mL isopropyl alcohol) was added (see Table 1), it reacted for 2 to 5 h. After the reaction was completed, the flask was cooled to room temperature in cold water, and the pH of the solution was adjusted to 7 with 1 M HCl, then transferred to a 3000 Da dialysis bag for 3 d of dialysis. The reaction solution was evaporated and concentrated at 50 °C. Five times the volume of ethanol was added, left standing overnight, filtered, separated, and dried to obtain different degrees of carboxymethylated *R. alismatis* polysaccharides (RAP1, RAP2, and RAP3).

#### 2.2.5. Detection of –COOH Content

Conductometric titration [26] was used to detect the content of –COOH in polysaccharides. Then, 10 mg of RAPs were completely dissolved into 40 mL of distilled water, and the titration was stopped by slowly adding 5 M of NaOH solution under magnetic stirring until the conductivity rose at a constant rate. The volume and conductivity changes of the consumed NaOH solution were recorded, and the –COOH content was calculated according to the following formula.
–COOH (%) = *C*_NaOH_ × (*V*_2_ − *V*_1_) × 45/(*C*_sample_ × *V*_sample_) × 100(1)
where *C*_NaOH_ is the molar concentration of NaOH solution (mol/L), *C*_sample_ is the concentration of polysaccharide solution (g/L), 45 is the molar mass of –COOH (g/mol), and *V*_sample_ is the volume of polysaccharide solution (mL). The conductometric titration curve can be divided into three sections: the stage of significant decrease in conductivity (A), the stage of equilibrium (B), and the stage of significant increase (C). Three tangent lines are drawn for these three stages, and the intersection points of A and B are *V*_1_; the intersection points of C with B is *V*_2_. This was repeated three times and the average value was obtained.

#### 2.2.6. FT-IR Characterization of RAPs

Completely dried RAPs (2 mg) and KBr (200 mg) were weighed and evenly mixed, ground, and pressed. The pressed tablet was placed into the instrument and scanned using an infrared light source in the range of 4000–400 cm^−1^ wave number with a resolution of 4 cm^−1^.

#### 2.2.7. ^1^H NMR and ^13^C NMR Characterization of RAPs

Completely dried RAPs (45 mg) were added to 0.6 mL of deuterium water (D_2_O). The mixture was completely dissolved and transferred to a nuclear magnetic tube. The structure of RAPs was detected by a 600-MHz nuclear magnetic resonance instrument under normal temperature and pressure.

#### 2.2.8. Zeta Potential Detection of RAPs

First, 2 mg of RAPs were dissolved in 10 mL of ultrapure water to prepare 0.20 mg/mL polysaccharide solution. Then, it was placed in the sample pool of Zeta potential analyzer for testing.

### 2.3. Antioxidative Activity of RAPs

#### 2.3.1. Ability to Scavenge ·OH Radicals

The ability of RAPs to scavenge •OH was detected by the H_2_O_2_/Fe system method [28]. The FeSO_4_ solution (1 mL, 2.5 mM) and o-phenanthroline solution (1 mL, 2.5 mM) were added to the test tube, then phosphate-buffered saline (PBS, 1 mL, pH = 7.4), H_2_O_2_ (1 mL, 20 mM), and 1 mL of RAPs solution (1, 2, 4, 8, and 10 mg/mL, respectively) were added successively. It was mixed and incubated at 37 °C for 90 min. The absorbance was measured at 536 nm and repeated three times to obtain the average absorbance (A3). Ascorbic acid was used instead of polysaccharide solution as a positive control group.
•OH removal rate% = (A3 − A1)/(A2 − A1) × 100(2)
where A1 is the absorbance of the background group, including 1 mL of H_2_O_2_, 1 mL of FeSO_4_, 1 mL of o-diazepine solution, 1 mL of PBS, and 1 mL of distilled water; A2 is the absorbance of the blank group, including 1 mL of FeSO_4_, 1 mL of o-diazepine solution, 1 mL of PBS, and 2 mL of distilled water.

#### 2.3.2. Ability to Scavenge DPPH Radicals

Each RAP solution (3 mL) (concentrations of 1, 2, 4, 8, and 10 mg/mL, respectively) and DPPH solution (1 mL, 0.4 mM, solvent: absolute ethanol) were mixed in a test tube. The final concentration of DPPH was 0.1 mM; the reaction was performed at 25 °C for 30 min in the dark, and the absorbance was measured at 517 nm. Ascorbic acid was used as a positive control group.
DPPH removal rate% = [1 − (A2 − A1)/A0] × 100(3)
where A0 was the detection absorbance of 3 mL of distilled water and 1 mL of DPPH solution mixture; A1 was the detection absorbance of 3 mL of polysaccharide sample solution and 1 mL absolute ethanol mixture; A1 was the detection absorbance of 3 mL of polysaccharide sample solution and 1 mL of DPPH solution mixture.

### 2.4. RAPs Regulate CaOx Crystal Growth

#### 2.4.1. Crystal Growth and Detection of Soluble Ca^2+^ Ions in the Supernatant

CaCl_2_ solution (20 mL, 22 mM) and a certain number of RAPs were added to a set of beakers, and distilled water was added to 24 mL [28]. After magnetic stirring for 5 min, Na_2_Ox solution (20 mL, 22 mM) was added to make the final volume 44 mL, *c* (Ca^2+^) = *c* (Ox^2−^) = 10 mM in the solution, and the final polysaccharide concentration was 0.04, 0.08, 0.12, 0.15, 0.20, and 0.30 mg/mL. After stirring the reaction for 10 min, it stood at 37 °C for 2 h and was then centrifuged. The obtained precipitate was washed three times with ethanol and dried to obtain CaOx crystals; the concentration of soluble Ca^2+^ ions in the supernatant was determined by ICP.

#### 2.4.2. X-ray Diffraction (XRD) Characterization of CaOx Crystals

The CaOx crystals obtained above were ground and pressed for X-ray diffraction detection; the test conditions were CuKᾱ ray, graphite monochromator, 40 kV, 15 mA, scanning range 10–50°, scanning speed 8°/min, step width 0.02°/s for qualitative and quantitative analysis [28].

The relative percentage contents of COD in the CaOx were calculated through the *K* value method and according to the XRD patterns:(4)$$COD\;\%=\frac{I_{COD}}{I_{COD}+I_{COM}}\times 100$$
where *I*_COD_ and *I*_COM_ are the intensity of the main crystal plane (200) of COD and the main crystal plane ($\bar{1}01$) of COM, respectively.

#### 2.4.3. FT-IR Characterization of CaOx Crystals

The experiment was the same as Section 2.2.6. Completely dried CaOx crystals (2 mg) and KBr (200 mg) were weighed and evenly mixed, ground, and pressed. The pressed tablet was placed into the instrument and scanned using an infrared light source in the range of 4000–400 cm^−1^ wave number with a resolution of 4 cm^−1^.

#### 2.4.4. Zeta Potential Detection of CaOx Crystals

CaOx crystals (5 mg) were added to 20 mL of pure water and sonicated for 5 min at low power. The liquid was then placed in the sample pool of zeta potential analyzer for detection. The liquid was shaken to keep it even before testing.

#### 2.4.5. Thermogravimetric Analysis (TGA) of CaOx Crystals

The thermograys analyzer was preheated for 30 min. CaOx crystals (5–10 mg) were taken and poured into a ceramic crucible, and then the crucible was put into the sample pool. Set the test parameters and start running the instrument. Nitrogen flow is adopted, and the heating rate is 10°/min. Thermogravimetric analysis curves of CaOx crystals were measured in the range of ~35–900 °C.

#### 2.4.6. Scanning Electron Microscope (SEM) Observation of CaOx Crystals

The above crystals (1 mg) were dispersed in 5 mL of anhydrous ethanol, and after 3 min of low-power ultrasound, the samples were spotted on a 10 mm × 10 mm glass slide and dried in an oven at 60 °C. After the samples were sprayed with gold, the size and shape of the crystal were observed with a field-emission scanning electron microscope.

#### 2.4.7. Nano Measurer (v1.2.5) Software to Analyze CaOx Crystal Size

Use Nano Measurer (v1.2.5) software to detect the crystal particle diameter in the SEM picture. The number of detected particles is 100. The software gives a summary table of crystal size. The picture was plotted using Origin software, and the average crystal size was obtained using a normal distribution function.

### 2.5. Interaction between FITC-Labeled Polysaccharides and Crystals

#### 2.5.1. Preparation and Characterization of FITC-RAPD

RAPD (100 mg) was dissolved in 5 mL of pure water. The pH of the solution was adjusted to 8 with 0.5 mol/mL NaHCO_3_; 5 mg of FITC was added, and the solution was stirred at room temperature in the dark for 24 h [38]. After the reaction is completed, absolute ethanol is filtered and added to the filtrate until no more precipitation is precipitated; the precipitate is then centrifuged and washed with absolute ethanol until there is no fluorescence absorption in the supernatant. The precipitate was dried to obtain FITC-RAPD, and it was characterized by FT-IR.

#### 2.5.2. FITC-RAPD Regulates CaOx Crystal Growth and Crystal Characterization

The experiment was the same as in Section 2.4.1. The formation of CaOx crystals was induced by the addition of FITC-RAPD and FITC at a final concentration of 0.1 mg/mL according to the reported method [39]. The above crystals (1 mg) were dispersed in 5 mL of absolute ethanol. After ultrasonication for 10 min, they were added dropwise to the well plates. After drying, the fluorescence expression of the crystals was observed with a fluorescence microscope.

### 2.6. Cytotoxicity of Crystals Regulated by RAPs

#### 2.6.1. Cell Culture

HK-2 cells were cultured in DMEM-F12 containing 10% FBS in a 5% CO_2_ incubator at 37 °C. The cells were passaged by trypsin digestion. In all experiments, the cells were cultured in seed plates for 24 h to reach 80% confluence before processing.

#### 2.6.2. Cell Viability Test

The cells were seeded in 96-well culture plates at 1.0 × 10^5^ cells/mL and 100 μL/well, incubated for 24 h, and then incubated with serum-free DMEM medium for 12 h to synchronize the cells. The medium was aspirated and washed twice with PBS. The cells were divided into two groups: (A) Blank control group: added serum-free medium; (B) Crystal group: added 300 μg/mL crystals for 6 h. After 6 h of incubation, added 10 μL of CCK-8 reagent to each well, incubated for 1.5 h at 37 °C, and measured the absorbance (A) at 450 nm with a microplate reader across five parallel groups; the average value was taken.
(5)$$Cell\;viability\;\%=\frac{A_{treatment\;group}}{A_{control\;group}}\times 100$$

#### 2.6.3. Lactate Dehydrogenase (LDH) Release Assay

Cells were treated as above and seeded in 96-well culture plates at a concentration of 1.0 × 10^5^ cells/mL and 200 μL/well. The culture wells were divided into four groups: (A) Background blank control wells: culture medium wells without cells; (B) Sample control wells: normal cell wells without crystal treatment; (C) Crystal damage wells: added 300 μg/mL crystals for 6 h; (D) Wells with maximum enzyme activity: wells of cells that were not treated with crystals for subsequent lysis. Subsequent experiments were performed in strict accordance with the instructions of the LDH kit, and the absorbance was measured at 490 nm.
(6)$$LDH\;\%=\frac{A_{group\;B/C}-A_{group\;A}}{A_{group\;D}-A_{grouo\;A}}\times 100$$

#### 2.6.4. Calcein/Propidium Iodide (PI) Staining Method to Detect Live and Dead Cells

The cells were seeded in 12-well culture plates at 1.0 × 10^5^ cells/mL and 1 mL/well. The experimental groups were the same as those in Section 2.6.2. After reaching the action time, the cells were washed twice with PBS; 0.5 mL of calcein-AM/PI detection solution was added to each well and incubated in the dark for 30 min at 37 °C. Then, the plate was observed by the inverted fluorescence microscope.

#### 2.6.5. ROS Level Detection

The cell seed plate and experimental groups were the same as those described in Section 2.6.4. After reaching the action time, 0.5 mL of DCFH-DA diluted was added, which was then incubated in the incubator at 37 °C for 30 min in the dark, the plate was observed by the inverted fluorescence microscope.

### 2.7. Statistical Analysis

Data were expressed as the mean ± SD. The experimental results were statistically analyzed by IBM SPSS Statistics 26 software, and the differences between the means of each experimental group and the control group were analyzed by Tukey test. *p* < 0.05 indicates a significance difference; *p* < 0.01, indicates a very significant difference; *p* > 0.05 means no significant difference.

## 3. Results

### 3.1. Carboxymethylation and Characterization of RAPs

#### 3.1.1. Separation, Purification, Degradation, and Monosaccharide Component Analysis of RAPs

Natural RAP was obtained through hot water extraction, while proteins were removed by the Sevag method and ethanol precipitation. RAP was degraded with 4% H_2_O_2_ for 2 h to obtain the degraded polysaccharide (RAPD).

A monosaccharide analysis of RAPD was performed by high-performance anion-exchange chromatography. The peak times of monosaccharide standard and RAPD are shown in Figure 1. RAPD showed only a single peak (Figure 1B) and was found to be a homogeneous polysaccharide mainly composed of glucose (98.3%), which was consistent with the literature [40].

#### 3.1.2. Carboxymethylation of RAPD and Detection of –COOH Content

RAPD was carboxymethylated by the chloroacetic acid solvent method. RAPs with different degrees of carboxymethylation were obtained by changing the concentration of chloroacetic acid and the reaction time (Table 1). When the reaction temperature was less than 70 °C, no side reactions occurred, and the chemical structure of the reactants was not easily destroyed [26].

The –COOH content in RAPs was detected by NaOH solution-conductometric titration, and the results are shown in Table 1. The –COOH contents of RAPD and three different carboxymethylated polysaccharides, RAP1, RAP2, and RAP3, were 3.57%, 7.79%, 10.84%, and 15.33%, respectively. Therefore, increasing the concentration of chloroacetic acid or the reaction time can increase the –COOH content in the polysaccharide.

#### 3.1.3. FT-IR Characterization of RAPs

The FT-IR spectra of RAPs were similar (Figure 2A), indicating that carboxymethylation did not have a large influence on the overall structure of the polysaccharides. Therefore, increasing the –COOH content of the polysaccharides did not change the overall structure of the polysaccharides. The characteristic peaks of RAPs at 3600–3200, 3000–2800, and 1250–1000 cm^−1^ were similar in height to the characteristic absorption peaks of dextran, indicating that the main structure and functional groups of RAPs are similar to those of dextran [41].

In the FT-IR spectra of RAPs, the strong and broad absorption peak at 3421 cm^−1^ was attributed to the stretching vibration of O–H in polysaccharide. The signal peak near 2927 cm^−1^ was attributed to the asymmetric stretching vibration of C–H in the polysaccharide backbone [42].

The absorption peaks observed around 1620–1604, 1421–1415, and 1326 cm^−1^ were attributed to the asymmetric stretching vibration of C=O, stretching vibration of C–O, and rocking vibration of –CH_2_–, respectively [41] (Table 2). In addition, a characteristic band appearing near 850 cm^−1^ in the fingerprint region indicated that the anomeric carbon of RAPs exhibited an α configuration [43].

FT-IR spectroscopy revealed the same content of polysaccharides in the samples (2 mg of RAPs and 200 mg of KBr). Therefore, the intensity of characteristic absorption peaks could reflect the content of functional groups in RAPs. As shown in Figure 2A, the absorption peak signals at 1602, 1420, and 1326 cm^−1^ gradually increased from RAPD to RAP1, RAP2, and RAP3, indicating the successful carboxymethylation of RAPD [43]. Owing to the gradual increase in the –CH_2_COOH group in the polysaccharide after carboxymethylation, its characteristic absorption peak was continuously enhanced.

#### 3.1.4. ^1^H NMR and ^13^C NMR Characterization of RAPs

According to the monosaccharide composition analysis of RAPD and the FT-IR characterization of RAPs, glucose was the main monosaccharide of RAPs (98.3%). Two main signal peaks appeared in the anomeric region (*δ* 4.4–5.6 ppm) of the ^1^H NMR spectrum of RAPD (Figure 2B), indicating that RAPD is mainly composed of two kinds of glucosyl residues [44], the main and side chains.

In the ^13^C NMR spectrum of RAPD, the signal peaks of C-1, C-2, C-3, C-4, C-5, and C-6 of the main chain appeared at *δ* 99.6, 71.5, 73.3, 76.7, 71.5, and 60.4 ppm (Table 3), respectively (Figure 2C), and the chemical shift of C-4 appeared in the downfield (*δ* 76.7 ppm) [45]; the main chain is →4)-α-D-Glc p-(1→). In the ^1^H NMR spectrum of RAPD, the signal peaks of *δ* 5.32, 3.56, 3.75, 3.57, 3.88, 3.77, and 3.67 ppm were attributed to the H-1-6 of the main chain [46]. The C-1-6 of the side chain residues appeared at *δ* 98.5, 72.9, 73.6, 71.4, 73.4, and 60.8 ppm; and H-1-6 appeared in sequence at *δ* 4.89, 3.53, 3.66, 3.34, 3.92, and 3.89 ppm. According to the proton of the residue and the chemical shift of carbon [47], this residue belongs to α-D-Glcp-(1→).

Comparison of ^13^C NMR spectra between RAPD and RAP3 showed that the signal peak at *δ* 60.1 ppm was weakened and that at *δ* 69.5 ppm was increased, indicating that the carboxymethyl group was substituted for the position of C-6. In addition, two new signal peaks appeared in the ^13^C NMR spectrum of RAP3. Among these two peaks, the new peak appearing near δ 80 ppm indicated that the carboxymethyl substitution might also occur at the C-4 position [48]. Meanwhile, another new peak near δ 178 ppm belonged to the carbonyl group introduced by carboxymethylation, and the signal peak at *δ* 72.6 ppm belonged to the carbon atom of the methylene group in the carboxymethyl group [49].

In summary, the main monosaccharide component of RAPs is glucose, and the main chain is →4)-α-D-Glcp-(1→; the linking residue is α-D-Glcp-(1→). The carboxymethylation modification of RAPD is successful, and the carboxymethyl substitution may occur at the C-4 and C-6 positions (Figure 2E).

#### 3.1.5. Zeta Potential Detection of RAPs

The zeta potential changes of the four RAP solutions are shown in Figure 2D. At 0.20 mg/mL, the zeta potential of RAPD was −36.3 mV. With the increase in –COOH content in the polysaccharide, the zeta potential of RAPs became negative. A large absolute zeta potential was associated with the strong dispersibility of RAPs in solution and the high-water solubility.

### 3.2. Antioxidative Activity of RAPs

The ability of the four RAPs to scavenge ·OH radicals and DPPH radicals is shown in Figure 3. With the increase in –COOH percentage in RAPs, their ability to scavenge free radicals was correspondingly enhanced.

At 10 mg/mL, the abilities of RAPD and RAP3 to scavenge ·OH radicals were 20.68% and 46.52%, respectively, and their abilities to scavenge DPPH radicals were 21.28% and 42.67%, respectively. In the concentration range of 1–10 mg/mL, RAPs showed a concentration-dependent scavenging ability for both free radicals. That is, when the concentration of RAPs increased, their ability to scavenge ·OH and DPPH radicals also increased.

### 3.3. RAPs Regulate CaOx Crystal Growth

#### 3.3.1. X-ray Diffraction Characterization

Figure 4 shows the XRD spectra of CaOx crystals regulated by different concentrations of RAPs. The blank control group, without polysaccharide and RAPD at various concentrations, only induced COM crystal formation (Figure 4A). The diffraction peaks at 14.88°, 24.2°, 30.16°, and 38.24° in 2θ were assigned to the ($\bar{1}01$), (020), ($\overset{-}{2}02$), and (130) planes of COM, respectively [28].

RAP1, RAP2, and RAP3 all induced the formation of COD crystals (Figure 4B–D), and the percentage of COD in the crystals increased with the –COOH content or polysaccharide concentration (Figure 4E). High concentrations of RAP2 and RAP3 induced 100% COD formation. The 2θ of COD crystals diffraction peaks appearing at 14.28°, 20.12°, 32.28°, and 40.28° were assigned to the (200), (211), (411), and (213) planes of COD, respectively [28].

When the concentration of RAP3 was increased from 0.12 mg/mL to 0.20 and 0.30 mg/mL, the percentage of COD in the crystals also increased from 93.21% to 100% and 100%, respectively.

Although RAPD and low RAP concentrations only induced the formation of COM, the diffraction intensity ratio (*I*($\bar{1}01$)/*I*(010)) of the ($\bar{1}01$) crystal planes to the (020) crystal planes of COM increased with the RAPD concentration. Similarly, at low concentrations (0.04 mg/mL), the *I*($\bar{1}01$)/*I*(010) of the COM induced by RAPs increased gradually with the increase in their –COOH content (Table 4).

#### 3.3.2. SEM Characterization

Given that the content of COD in the controlled crystals increased gradually when the concentration of the four RAPs was 0.12 mg/mL, the crystals with this concentration were selected for SEM observation, as shown in Figure 5A.

The blank group without polysaccharide and the RAPD group only formed COM crystals that had sharp edges and irregular morphology and seriously aggregated. The difference is that more gaps can be found between the COM crystals of the RAPD group compared with those in the blank. When RAP1, RAP2, and RAP3 were added, round and blunt quadrangular bipyramidal COD crystals appeared. With the increase in –COOH content in the polysaccharide, the percentage of COD in the crystals also increased, the crystals edges became rounded and blunt, and the dispersibility of the crystals was significantly enhanced.

The RAP3 with the highest content of –COOH was selected, and its concentration was continuously increased. The obtained crystals were almost all pure COD (Figure 5A), the thickness of the COD crystals increased, the typical quadrangular bipyramid was bright, and the crystals size decreased. Detection using Nano Measurer (v1.2.5) software showed that the COD crystal sizes induced by RAP3 at 0.12, 0.20, and 0.30 mg/mL were 700, 634, and 457 nm, respectively (Figure 5C).

#### 3.3.3. FT-IR Characterization

FT-IR spectra were used to characterize the CaOx crystals induced by RAPs at different concentrations (Figure 6). The FT-IR spectra of the crystals in the blank group and RAPD group (Figure 6A) showed, which corresponded to the in-plane O–C=O bending, out-of-plane O–H deformation, and CO_2_ wagging mode, respectively [50].

The FT-IR spectra of CaOx crystals induced by low concentrations of RAP1, RAP2, and RAP3 (Figure 6B–D) were similar to those of the RAPD group, indicating that the crystals were dominated by COM. However, the high concentrations of RAPs mainly showed the characteristic peaks of COD. That is, only one strong and broad absorption peak appeared in the region of 3488–3060 cm^−1^; the asymmetric and symmetrical stretching vibration peaks of C=O gradually blue-shifted to around 1640 and 1328 cm^−1^, respectively; and the characteristic peaks of COD in the fingerprint region were around 917 and 620 cm^−1^ [51].

#### 3.3.4. Zeta Potential

Figure 7A shows the zeta potential of CaOx crystals regulated by RAPs with different –COOH contents. The zeta potential of the blank crystal was −2.79 mV, and that of the crystals induced by each RAP increased with RAP concentration and –COOH content. The absolute zeta potentials of the crystals increased, and the effect of –COOH content was greater than that of RAP concentration.

Crystals with a large absolute zeta potential have a great mutual repulsion force in solution, and their suspensions are highly stable [52]. CaOx crystals that do not easily settle are beneficial to reduce the risk of CaOx kidney stone formation [53].

#### 3.3.5. Soluble Ca^2+^ Concentration in the Supernatant

Figure 7B shows the Ca^2+^ concentration in the supernatant after crystal formation induced by the four RAPs at 0.12 mg/mL. When the –COOH content in RAPs was high, the Ca^2+^ chelation ability in the system was strong, the supersaturation of the system was low, and the soluble Ca^2+^ concentration was high. This effect was intensified by the increase in the –COOH content of RAPs.

### 3.4. Interaction between RAPs and CaOx Crystals

#### 3.4.1. TGA Characterization

The standard CaOx crystal decomposition is divided into three steps [54], and the decomposition reaction equation is as follows:(1)CaC_2_O_4_.nH_2_O → CaC_2_O_4_ + nH_2_O (n = 1 or 2)(2)CaC_2_O_4_ → CaCO_3_ + CO(3)CaCO_3_ → CaO + CO_2_

The TGA curves of CaOx crystals regulated by each RAP are shown in Figure 8A. The polysaccharide-regulated group crystals can be divided into four weightless processes corresponding to W1, W2, W3, and W4 in Figure 8A. The first weight loss was completed before the temperature reached 230 °C and was attributed to the decomposition of crystal water. When the temperature continued to increase to 450 °C, a small proportion of weight loss occurred (W2) due to the decomposition of RAPs mixed into the crystal [55]. When the temperature increased to 520 °C, the crystal decomposed for the third time (W3), CaOx was decomposed into CaCO_3_ and CO, and the escape of CO caused the weight loss [54]. When the temperature was continued to increase to 750 °C, the crystals decomposed for the last time, CaCO_3_ was decomposed into CaO and CO_2_, and the escape of CO_2_ caused the fourth weight loss (W4).

The weight loss data of each crystal are shown in Table 5. The weight loss of the blank group crystal without polysaccharide was 12.17%, 18.12%, and 29.69% (Table 5), which were almost consistent with the standard COM theoretical weight loss percentages (12.3%, 19.2%, and 30.1%) [54]. The first weight loss (12.03%) and decomposition temperature (193.69 °C) of the RAPD group crystals also corresponded to the loss of one water molecule from COM, indicating that the crystals are mainly COM crystals. These findings are consistent with the results of XRD and FT-IR.

With the increase in the –COOH content of RAPs, the percentage of COD in the crystals also increased (Figure 4E). COD crystals had one additional molecule of crystal water than COM. Hence, the first weight loss percentage of RAP3-regulated crystals with the highest COD content was the largest (18.98%). In addition, the first weight loss of the RAPD-regulated crystals was slightly lower than that of the blank because the RAPD-regulated crystals were mixed with polysaccharides, and their actual COM content was lower than that of the blank group. Hence, the content of crystal water was also lower in the RAPD-regulated crystals than in the blank group.

The RAP-regulated crystals differed from those in the blank group in the second weight loss (W2 in Figure 8A). According to the comparison of the derivative thermogravimetry (DTG) curves of the blank group crystals and the RAP3-regulated crystals (Figure 8B,C), the blank group crystals experienced only one mass loss at 400–500 °C, and the RAP3-regulated crystals experienced two mass losses at 400–500 °C. The loss (W2) at 430 °C was attributed to the decomposition of the adsorbed RAP3 in the crystal [56,57], and that at 475 °C was due to the third weight loss of the crystals [54]. Therefore, the polysaccharide-regulated crystals differed from those in the blank at the second weight loss. The differences among the groups of crystals were attributed to the adsorption of polysaccharides by the crystals [55].

#### 3.4.2. Adsorption of RAPs and Crystals

(a)FT-IR characterization of FITC-RAPD

The FT-IR spectra of RAPD before and after FITC labeling are shown in Figure 9A. Compared with RAPD, FITC-labeled RAPD (FITC-RAPD) showed a new signal at 1332 cm^−1^ (the blue arrow in Figure 9A). This signal was attributed to the amide group C=O bond in FITC [58], indicating that RAPD was successfully labeled by FITC.

(b)Fluorescence microscope observation of crystals under FITC-RAPD regulation

Figure 9B shows the fluorescence microscope images of CaOx crystals regulated by FITC-RAPD and FITC. The crystals regulated by FITC alone did not exhibit green fluorescence, indicating that FITC fails to interact with CaOx crystals. Almost all of the CaOx crystals regulated by FITC-RAPD exhibited green fluorescence, and the brightfield images (i.e., crystals) completely overlapped with the fluorescent images (i.e., fluorescent-labeled polysaccharides). This finding indicated an interaction between RAPD and CaOx crystals, which is consistent with the results of thermogravimetric analysis (Figure 8).

### 3.5. Cytotoxicity of RAP-Regulated Crystals

#### 3.5.1. Cell Viability

The cytotoxicity of each group of crystals is shown in Figure 10A. The CaOx crystals formed without polysaccharides (crystal 1 in Figure 10) were the most cytotoxic, and the cell viability was only 42.75% of the blank control group. The cytotoxicity of crystals in the RAPD, RAP1, RAP2, and RAP3 control groups (crystals 1–4 in Figure 10) at a concentration of 0.12 mg/mL decreased sequentially. When the –COOH in the RAPs was high, the cytotoxicity of the crystals was low. This phenomenon was attributed to the increased content of COD in the crystals and the decreased degree of crystal aggregation. Given that COD is less cytotoxic than COM, the aggregated crystals are also more cytotoxic than the dispersed crystals. When the concentration of RAP3 was increased from 0.12 to 0.20 and 0.30 mg/mL, the cytotoxicity of the crystals (crystals 5–7 in Figure 10) increased mainly due to the gradual decrease in crystal size.

#### 3.5.2. LDH Release Amount

LDH is a stable cytoplasmic enzyme present in the cytoplasm of all tissue cells. Once the cell membrane is damaged and ruptured, LDH is released outside the cell. Therefore, the amount of released LDH can be used as a marker of cell membrane integrity. A high amount of released LDH indicates a high degree of damage to the cells [59].

Figure 10B shows the effects of the crystals in each group on the release of LDH from HK-2 cells. The crystals in the blank group had the most serious damage to the cell. Meanwhile, the damage ability of the crystals to the cell membrane in each RAP-regulated group was reduced. However, high RAP3 concentrations showed great damage capacity due to crystal size reduction. This finding is consistent with the results of cell viability experiments.

#### 3.5.3. Calcein/PI Staining to Detect Cell Death

The cell death state induced by each group of crystals was detected by calcein (calcein-AM) and PI staining (Figure 11). Calcein-AM is a fluorescent marker dye for living cells that does not emit fluorescence. After entering living cells, it is cleaved by intracellular esterase to form a membrane-impermeable polar molecule, calcein, which is retained in the cell and emits strong green fluorescence. PI cannot pass through the cell membrane of living cells but can pass through the disordered region of the dead cell membrane to reach the nucleus where it embeds in the DNA double helix of the cell to produce red fluorescence. Therefore, PI can only dye dead cells red.

The cells in the blank control group were closely linked, and the cell density was large. No red fluorescence appeared, indicating that all the cells were alive. After the blank group crystals were added, the cells showed strong red fluorescence, indicating that cell death occurred. In addition, the cell density decreased because the cells no longer adhered after death and were removed during staining [60]. For the crystals of each RAP group, a decrease in the degree of cell death and a gradually increase in the cell density were observed with the increase in the –COOH content of RAPs (Figure 11). For the crystals regulated by different concentrations of RAP3, their cytotoxicity increased with RAP3 concentration due to the decreasing size of COD crystals.

#### 3.5.4. ROS

The changes in cellular ROS expression caused by RAP-induced CaOx crystal formation are shown in Figure 12. The green fluorescence of cells in the blank control group without crystals was extremely weak, indicating that the expression of ROS was extremely low. The blank group crystals induced high ROS levels in the cells. The intensities of cellular ROS induced by the crystals in each RAP group were lower than those in the blank group crystals (1 in Figure 12B). With the increase in the –COOH content of RAPs, the expression of ROS decreased. The ROS intensity induced by the crystals regulated by RAP3 was enhanced with the increasing RAP3 concentration (Figure 12).

## 4. Discussion

### 4.1. Carboxymethylation and Characterization of RAPs

Carboxymethylation modification is one of the effective means to improve the biological activity of natural plant polysaccharides [24,25,26]. In this work, three modified polysaccharides, RAP1, RAP2, and RAP3, with –COOH contents of 7.79%, 10.84%, and 15.33%, respectively, were obtained after the carboxymethylation of RAPD with –COOH content of 3.57% by the chloroacetic acid solvent method.

Monosaccharide analysis showed that RAPD was mainly composed of glucose. FT-IR, ^1^H NMR, and ^13^C NMR spectra showed that the overall structure of carboxymethylated RAPs did not change upon modification. The main chain was →4)-α-D-Glcp-(1→), which contained part of the residue α-D-Glcp-(1→.

FT-IR spectra showed that, compared with RAPD, the carboxymethylated RAPs showed significant enhancement in the signal peaks around 1604, 1421, and 1326 cm^−1^ corresponding to the asymmetric stretching vibration of C=O, stretching vibration of C–O, and swing vibration of –CH_2_– in –CH_2_COOH, respectively.

In the ^13^C NMR spectrum, a new signal peak appeared around *δ* 178 ppm, which was attributed to the C=O in the introduced carboxymethyl group (–CH_2_COOH). The intensity of the new signal increased with the –COOH content of RAPs.

### 4.2. Enhanced Antioxidant Capacity of RAPs

The carboxymethyl-modified RAPs showed a strong ability to scavenge ·OH radicals and DPPH radicals (Figure 3). The biological activity of plant polysaccharides is closely related to their active group content (such as –OH, –COOH, –OSO_3_^−^, and –OPO_3_H_2_) [22]. Carboxymethylation modification obviously increased the antioxidant activity and, hence, the biological activity of *Sargassum fusiforme* polysaccharide [61], carboxymethyl xylan polysaccharide [25], and *Ganoderma lucidum* polysaccharides [48]. Water solubility is an important factor to promote the biological activity of polysaccharides. Zhang et al. [62] reported that carboxymethylated derivatives of *Pholiota nameko* polysaccharides showed better water solubility and antioxidant capacity. After the carboxymethylation of RAPD, additional hydrophilic groups –COOH were introduced and significantly improved its water solubility and antioxidant activity [26]. In addition, the introduced –COOH can inhibit the generation of ·OH radicals by chelating Fe^2+^ ions in the system, and the generated acidic environment will weaken the interaction of hydrogen bonds on the polysaccharide structure, promote the breaking of –OH bonds on the sugar ring, capture additional free radical, and, thus, further enhance the antioxidant activity of RAPs [10,48].

### 4.3. Ability of RAPs to Regulate the Growth of CaOx Crystals

#### 4.3.1. Induce COD Formation and Inhibit COM Growth

FT-IR, XRD, SEM, and other tests showed that the high –COOH content or RAP concentration was associated with low COM percentage in the induced crystals, high COD percentage, and small crystal size. When the abundant –COOH in RAPs chelated Ca^2+^ ions, the Ca^2+^ concentration around the polysaccharide increased, resulting in a high-energy interface [10]. The [Ca^2+^]/[Ox^2−^] ratio of the local space around the polysaccharide was also significantly increased, which is favorable for the formation of COD [63]. After RAPs chelated Ca^2+^ ions, the supersaturation of the solution decreased, which is also conducive to the formation of COD [11].

#### 4.3.2. Change the Strength of Different Crystal Planes of COM

Although RAPD only induced the formation of COM, the diffraction intensity ratio (*I*($\bar{1}01$)/*I*(010)) of the two main crystal planes ($\bar{1}01$) and (020) of the COM gradually increased with RAPD concentration (Figure 3F). Among the RAPs with different –COOH contents, the *I*($\bar{1}01$)/*I*(010) was 0.52, 0.64, 0.90, 0.91, and 0.98 for the blank, RAPD, RAP1, RAP2, and RAP3 groups, respectively (Table 4).

Sheng et al. [7] showed that the density of Ca^2+^ ions (sites/Å2) varies in the different crystal planes of COM and COD crystals: COM ($\bar{1}01$) = 0.0542 sites/Å^2^ > COD (100) = 0.0439 sites/Å^2^ > COM (010) = 0.0333 sites/Å^2^ > COD (101) = 0.0225 sites/Å^2^. RAPs are rich in –COOH groups, which preferentially bind to the relatively Ca^2+^-rich crystal plane ($\bar{1}01$) through electrostatic interaction and promote the growth of the crystal along the [010] direction, thus extending the crystal plane ($\bar{1}01$) [39,64]. The obtained COM crystal has a small acute angle at the edge and tip, and the *I*($\bar{1}01$)/*I*(010) also gradually increases (Figure 4G), which is conducive to the crystals passing through the urethra and reduces the risk of stones.

#### 4.3.3. Inhibition of Crystal Aggregation

Crystal aggregation increases the risk of crystal retention and promotes the formation of kidney stones [3,20]. RAPs can improve crystal dispersion in solution and inhibit crystal aggregation and reduce the size of crystals, which are easily excreted with urine [7]. When RAPs were adsorbed on the crystals surface, they made the zeta potential of the crystals negative by chelating Ca^2+^ ions. The more negative the zeta potential, the more stable the system. That is, the COM crystals repel each other, the crystal dispersion is high, and the aggregation is suppressed [12]. Viswanathan et al. [65] found that normal THP removes sialic acid residues (ds-THP) and promotes the aggregation of CaOx crystals due to its net anion reduction, which completely reverses the function of normal THP to inhibit crystal aggregation.

### 4.4. Reasons for Differences in the Cytotoxicity of CaOx Crystals Regulated by Different RAPs

#### 4.4.1. Reduced Cytotoxicity of Polysaccharide-Regulated Crystals

Urine microcrystals can damage cells, and damaged cells are likely to adhere to urine microcrystals. Crystals adhering to the surface of cells can also cause oxidative stress in cells, damaging various subcellular organelles and leading to cell necrosis. In addition, cell damage causes an imbalance of homeostasis. Additional adhesion molecules are expressed on the cell surface and cause the adherence of additional urine microcrystals; dead cell debris also bind to the crystals [66,67]. These vicious cycles aggravate the formation of kidney stones [27,33].

After CaOx crystals damaged the cells, cell viability decreased, and LDH release (Figure 10B) and ROS expression increased (Figure 12), resulting in an increase in the number of dead cells (Figure 11). Compared with that in the blank group crystals without polysaccharide, the cytotoxicity of the polysaccharide group crystals was reduced, and the degree of reduction was positively correlated with the –COOH content in the polysaccharide.

#### 4.4.2. Differences in Cytotoxicity between COM and COD

The carboxymethyl-modified RAPs were more effective than RAPD in inducing the formation of COD and inhibiting the growth of COM. Given that pure COM crystals are more cytotoxic than COD, the cytotoxicity gradually decreases when the COD content in the crystals increases.

Compared with COD, the incidence of COM stones is twice higher [8], its aggregation degree is often higher, and its adhesion to renal epithelial cells is also stronger. That is, COM stays in the body. By contrast, COD microcrystals are frequently found in the urine of asymptomatic individuals and are a relatively benign crystal form [68].

In the urine of the control group without kidney stones, many benign COD microcrystals were formed due to the interaction between urine macromolecules and urine microcrystals. These crystals have a small size and round shape, do not easily aggregate, and are easily flushed out of the body [65,66,67]. COD crystallites found in urine are often mixed with urine macromolecules, such as citrate, pyrophosphate, and GAGs [9]. Wesson et al. [69] showed that isolated urine macromolecules can convert COM to COD in vitro. In-vitro studies by Semangoen et al. [14] found no significant difference in death between the COD-treated cells and the normal group. By contrast, Vinaihat et al. [15] showed that COM massively aggregated and adhered to renal epithelial cells, inducing significant cell death.

#### 4.4.3. Small COD Size Leads to Great Cytotoxicity

Figure 10, Figure 11 and Figure 12 show that with the increase in RAP3 concentration from 0.12 mg/mL to 0.20 and 0.30 mg/mL, the size of the crystals decreased, and their cytotoxicity was enhanced. The percentages of COD in these crystals are 93.21%, 100%, and 100%. That is, the crystals were mainly composed of COD. The difference in cytotoxicity is mainly due to the crystal size: small COD size is associated to great cell toxicity, which is consistent with the findings of Sun et al. [70].

When crystals with small size adhered to cells, parts of the crystals were internalized into the cells. The continuous internalization of crystals can disrupt the intracellular homeostasis, trigger oxidative stress, and further lead to organelle disorders; partially internalized crystals may even enter the nucleus to damage DNA, which can seriously lead to cell death [71].

### 4.5. Interaction between RAPs and Crystals

Fluorescent labeling experiments (Figure 9) and TGA detection (Figure 8) showed that RAPs interact with CaOx crystals and could change the crystal shape, size, and surface charge to obtain COD crystals with low toxicity.

RAPs can be adsorbed on the surface of CaOx crystals or embedded in the crystals. TGA data showed that the contents of polysaccharides in the crystals regulated by RAPD, RAP1, RAP2, and RAP3 were 2.22%, 5.86%, 6.06% and 6.50%, respectively. After RAPs entered the crystals, the absolute zeta potential of the crystals increased, their growth rate decreased, and the aggregation between crystals was inhibited.

Gomes et al. [36] found that the interaction of negatively charged seaweeds’ sulfated polysaccharides with the crystal surface can increase the negative charge on the crystal surface, change the crystallization dynamics, greatly reduce the crystal size, generate rounded COD crystals, and allow COD to exist stably. Akın et al. [61] found that carboxymethyl inulin (CMI) containing carboxylic acid groups are adsorbed on the active growth sites on the surface of CaOx crystals, thereby inhibiting the growth of crystals; the ability of CMI to inhibit the growth of CaOx crystals is related to its anion content. Urine molecules such as GAGs [16] and potassium citrate [17] are also adsorbed on the surface of CaOx crystals through their own negative charges, thus changing the zeta potential of the crystal surface and the kinetics of the crystals, effectively inhibiting the aggregation between crystals and reducing the rate of crystal growth and risk of stones.

The mechanism of RAPs in regulating crystals is as follows: RAPs are rich in polyanions, such as –COOH, which can chelate a large amount of Ca^2+^ ions in the solution, reduce the supersaturation of the solution, inhibit the growth of COM, and induce the formation of COD. RAPs are also adsorbed on the surface of CaOx crystals, thus increasing the negative charge of the crystal surface. That is, increasing the absolute zeta potential and the mutual repulsive force between the crystals to inhibit the aggregation of the crystals (Figure 13). RAPs change the composition, size, morphology, and cytotoxicity of CaOx crystals; crystals with a high percentage of COD have low cytotoxicity. The cytotoxicity of pure COD crystals decreases with their size. The degree of interaction between RAPs and crystals is positively correlated with the –COOH content of RAPs.

## 5. Conclusions

The natural degradation polysaccharide RAPD (3.57%) was selected for carboxymethylation, and three modified polysaccharides RAPs with –COOH contents of 7.79%, 10.84%, and 15.33% were obtained. The biological activity of RAPs was positively correlated with their –COOH content. Carboxymethylated RAPs can induce the formation of COD, inhibit the growth of COM, increase the concentration of soluble Ca^2+^ ions in the system and the absolute value of zeta potential on the crystal surface, inhibit the aggregation of crystals, and result in the crystal edges and corners becoming rounded and blunt. All of these effects are beneficial to reduce the adhesion of crystals to cells and the risk of kidney stone formation. Different RAPs induced the formation of CaOx crystals with different cytotoxicity. The cytotoxicity of the crystals was reduced when the percentage of COD content in the crystals increased, the degree of aggregation of the crystals decreased, the shape of the crystals was rounded, or the number of RAPs adsorbed by the crystals increased. However, the decrease in the size of the crystals increased the cytotoxicity. Compared with CaOx crystals formed in the absence of polysaccharides, the cytotoxicity of RAP-regulated crystals was reduced. That is, the presence of RAPs could reduce the cytotoxicity of calcium oxalate crystals formed. The results showed that RAPs can reduce the damage of CaOx to HK-2 cells by optimizing the composition, size, and shape of CaOx crystals and effectively reduce the risk of kidney stone formation. RAPs, especially RAP3 with high carboxyl group content, may be potential drugs for the prevention and treatment of calcium oxalate kidney stones.

## Figures and Tables

**Figure 1 biomolecules-13-01044-f001:**
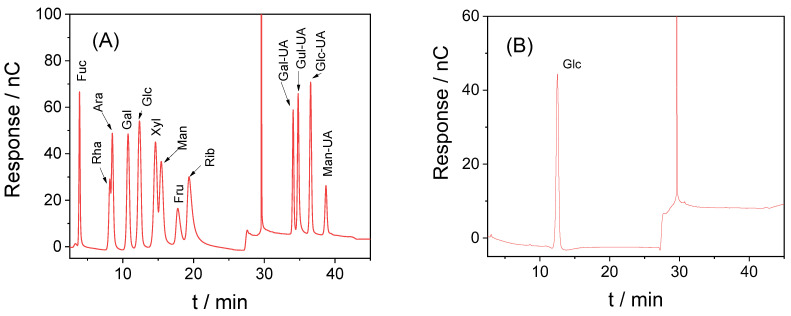
Monosaccharide composition analysis of RAPD. (**A**) Peak time and peak area of standards; (**B**) RAPD.Glc (tR: 12.359 min).

**Figure 2 biomolecules-13-01044-f002:**
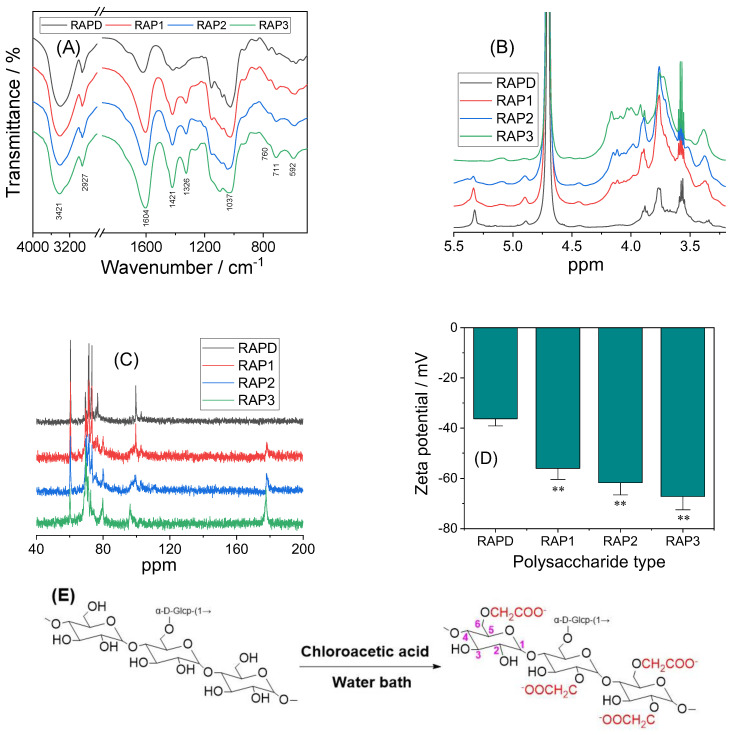
Analysis of RAPs. (**A**) FT-IR spectra; (**B**) ^1^H-NMR spectra; (**C**) ^13^C-NMR spectra; (**D**) Zeta potential at a concentration of 0.20 mg/mL, Compared with RAPD group; ** *p* < 0.01; (**E**) Structure and carboxymethylation equation of RAPs. The numbers represent the carbon number of the monosaccharide structure.

**Figure 3 biomolecules-13-01044-f003:**
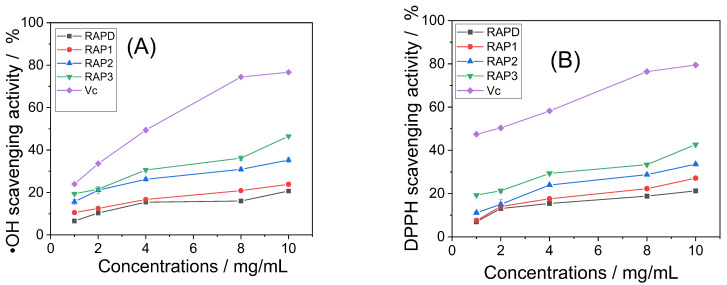
Changes in antioxidant capacity of RAPs. (**A**) Scavenging·OH radical; (**B**) Scavenging DPPH radical.

**Figure 4 biomolecules-13-01044-f004:**
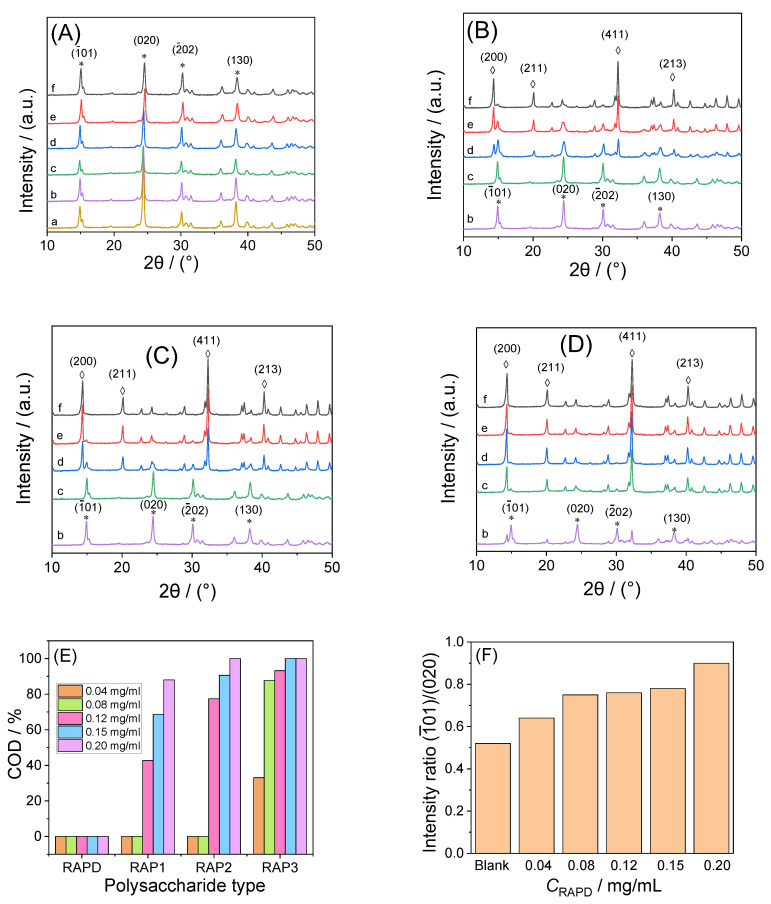
XRD spectra and COD content of CaOx crystals regulated by different concentrations of each RAP. (**A**) RAPD; (**B**) RAP1; (**C**) RAP2; (**D**) RAP3; (**E**) The percentage of COD in crystals; (**F**) The plane-strength ratio *I*($\bar{1}01$)/*I*(010) of COM crystals induced by different concentrations of RAPD; (**G**) The plane strength ratio *I*($\bar{1}01$)/*I*(010) of CaOx crystals induced by each RAPs at 0.04 mg/mL. c(RAPS): (a) 0; (b) 0.04; (c) 0.08; (d) 0.12; (e) 0.15; (f) 0.20 mg/mL. * Indicates the characteristic diffraction peaks of COM; ◊ Indicates the characteristic diffraction peaks of COD.

**Figure 5 biomolecules-13-01044-f005:**
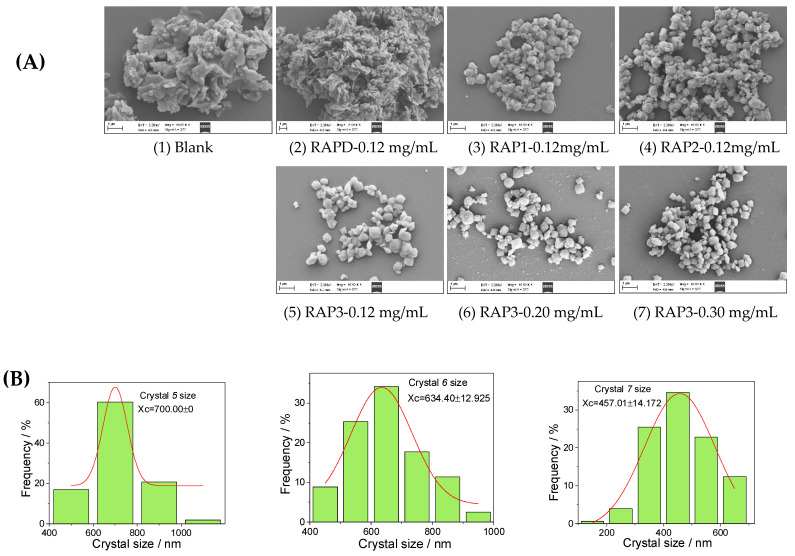
SEM image of CaOx crystals induced by (**A**) 0.12 mg/mL each RAPs and different concentrations of RAP3. (**B**) The average diameter of COD crystals induced by different concentrations of RAP3. SEM shooting conditions: EHT: 5.00 kV, Mag: 10.00 KX, WD: 4.6 mm, Signal A: SE2. Scale: 1 μm.

**Figure 6 biomolecules-13-01044-f006:**
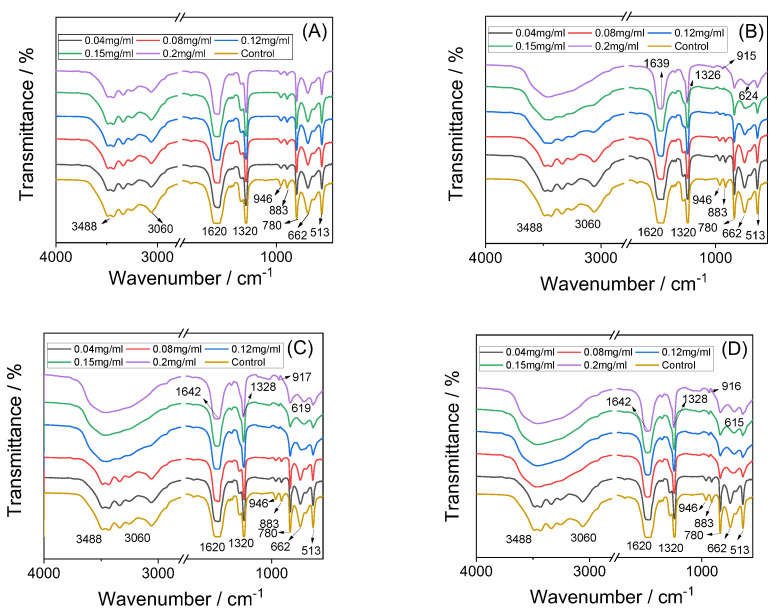
FT-IR spectra of CaOx crystals regulated by different concentrations of each RAPs. (**A**) RAPD; (**B**) RAP1; (**C**) RAP2; (**D**) RAP3.

**Figure 7 biomolecules-13-01044-f007:**
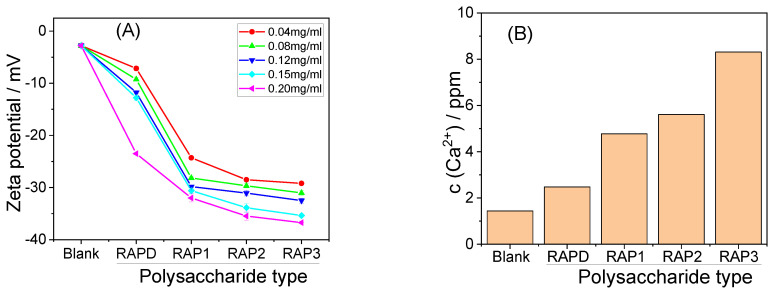
Zeta potential of CaOx crystals formed by different concentrations of each RAP (**A**) and soluble Ca^2+^ ion concentration in the supernatant when RAP concentration is at 0.12 mg/mL (**B**).

**Figure 8 biomolecules-13-01044-f008:**
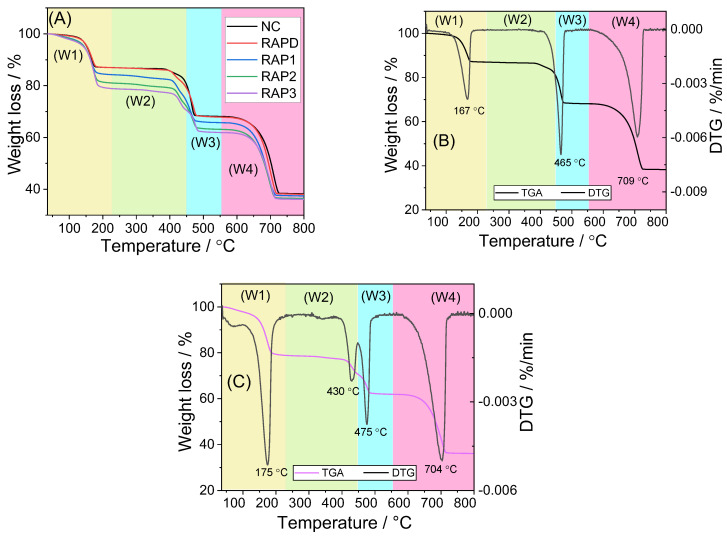
The TGA curves of CaOx crystals regulated by each RAPs at 0.12 mg/mL (**A**) and the TGA and DTG curves of blank group crystals (**B**) and RAP3-regulated crystals (**C**). W1-W4 (color blocks) are the temperature range of each time for the crystals to lose weight.

**Figure 9 biomolecules-13-01044-f009:**
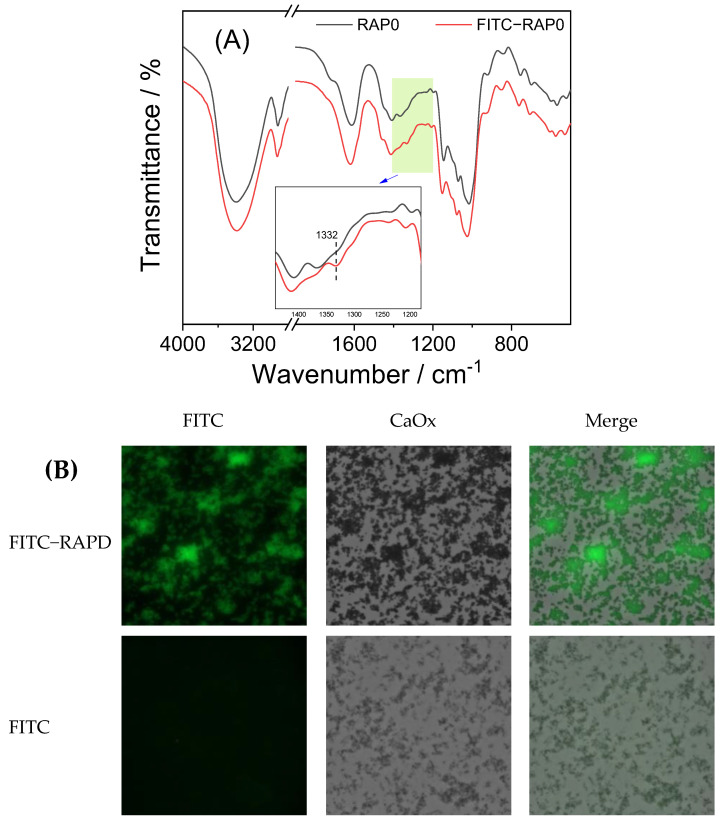
Interaction of FITC-RAPD with CaOx crystals. (**A**) FT-IR spectra of FITC-RAPD and RAPD; (**B**) Microscope image of CaOx crystals regulated by FITC-RAPD and FITC. The blue arrow represents the new signal peak of amide group (C=O bond) at 1332 cm^−1^.

**Figure 10 biomolecules-13-01044-f010:**
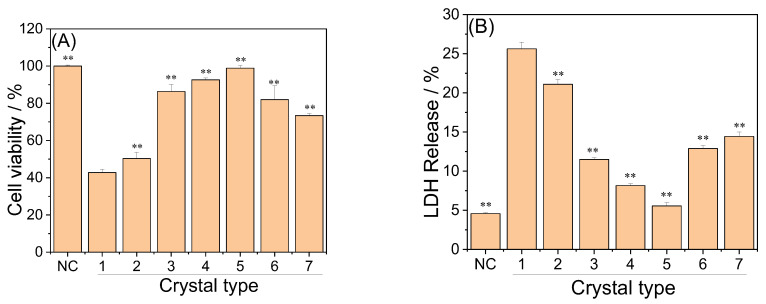
Toxicity of CaOx crystals induced by 0.12 mg/mL each RAPs and different concentrations of RAP3 to HK-2 cells. (**A**) Cell viability; (**B**) LDH release. NC: normal control group; Crystals group: CaOx (1–7) injury group, CaOx concentration: 300 μg/mL, injury time: 6 h. The crystal numbers are the same as in Figure 5. Compared with 1 group, ** *p* < 0.01.

**Figure 11 biomolecules-13-01044-f011:**
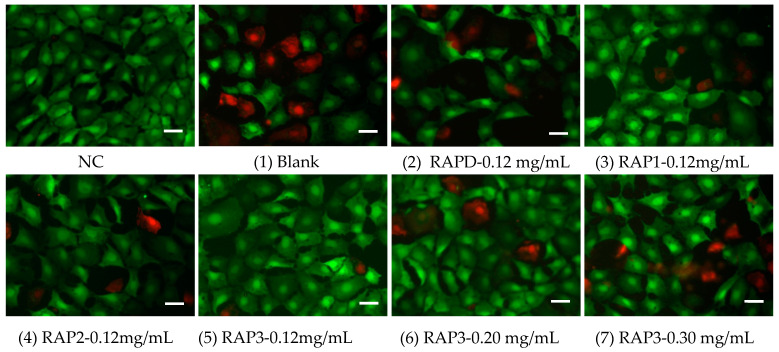
Analysis of cell death caused by CaOx crystals induced by 0.12 mg/mL each RAPs and different concentrations of RAP3 by calcein/PI staining. NC: normal control group; Crystals group: CaOx (1–7) injury group, CaOx concentration: 300 μg/mL, injury time: 6 h. Scale: 50 μm. The crystal numbers are the same as in Figure 5. Green fluorescence: live cells; Red fluorescence: dead cells.

**Figure 12 biomolecules-13-01044-f012:**
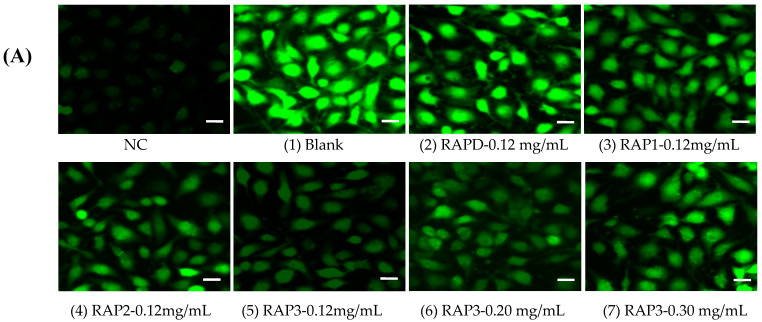
Cellular ROS expression induced by CaOx crystals induced by (**A**) 0.12 mg/mL each RAP and different concentrations of RAP3. (**B**) Fluorescence quantitative histogram. NC: normal control group; Crystals group: CaOx (1–7) injury group, CaOx concentration: 300 μg/mL, injury time: 6 h. Scale: 50 μm. The crystal numbers are the same as in Figure 5. Compared with 1 group, ** *p* < 0.01.

**Figure 13 biomolecules-13-01044-f013:**
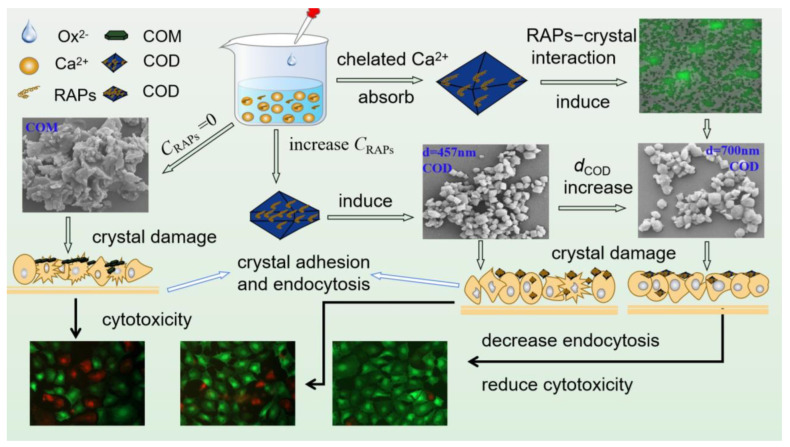
Mechanism of RAPs regulating calcium oxalate crystal growth and crystal cytotoxicity.

**Table 1 biomolecules-13-01044-t001:** Carboxymethylation conditions and carboxyl group content of *R. alismatis polysaccharides*.

Sample	Reaction Time /h	Chloroacetic Acid /g	Temperature Reflex /°C	–COOH Content /%
RAPD	/	/	/	3.57
RAP1	2	2.5	50	7.79
RAP2	4	3	60	10.84
RAP3	5	5	60	15.33

**Table 2 biomolecules-13-01044-t002:** The characteristic absorption peaks of FT-IR before and after the carboxymethylation of *R. alismatis* polysaccharides.

Sample	–COOH Content/%	Characteristic Absorption Peak/cm^−1^					
		O–H	C–H	C=O Asymmetric	C–O	–CH2– from Carboxymethyl	C–O–C
RAPD	3.57	3403	2927	1620	1415	/	1024
RAP1	7.79	3415	2927	1604	1419	1326	1026
RAP2	10.84	3413	2925	1604	1419	1326	1043
RAP3	15.33	3421	2927	1604	1421	1326	1037

**Table 3 biomolecules-13-01044-t003:** Chemical shifts (*δ*) in ^1^H NMR and ^13^C NMR spectra of RAPD and RAP3.

Sample	Residue	Chemical Shift (ppm)					
		C1/H1	C2/H2	C3/H3	C4/H4	C5/H5	C6/H6
RAPD	(1→4)-α-D-Glc*p*	99.6/5.32	71.5/3.56	73.3/3.75	76.7/3.57	71.5/3.88	60.4/3.77, 3.67
	α-D-Glc*p*-(1→)	98.5/4.89	72.9/3.53	73.6/3.66	71.4/3.34	73.4/3.92	60.8/3.89
RAP3	(1→4)-α-D-Glc*p*	96.2/5.70	71.0/3.58	73.3/3.75	76.6/3.58	71.0/3.91	60.1/3.90, 3.72
	α-D-Glc*p*-(1→)	92.0/4.91	72.5/3.57	73.3/3.60	71.4/3.38	73.4/4.03	60.3/4.00

**Table 4 biomolecules-13-01044-t004:** The crystal plane strength ratio *I*($\bar{1}01$/*I*(010) of COM crystals induced by different RAPD concentrations and different carboxymethylated RAPs.

Polysaccharide Types	Concentration /mg/mL	COD /%	$\mathrm{Crystal}\;\mathrm{Plane}\;\mathrm{Strength}\;\mathrm{Ratio}\;I(\bar{1}01)/I(010)$
Blank	/	0	0.52
RAPD	0.04	0	0.64
RAPD	0.08	0	0.75
RAPD	0.12	0	0.76
RAPD	0.15	0	0.78
RAPD	0.20	0	0.90
Blank	/	0	0.52
RAPD	0.04	0	0.64
RAP1	0.04	0	0.90
RAP2	0.04	0	0.91
RAP3	0.04	33.02	0.98

**Table 5 biomolecules-13-01044-t005:** Weight loss data of CaOx crystals induced by each RAPs at 0.12 mg/mL.

Polysac Charides	COD /%	W1 /%	T1 /°C	W2 * /%	T2 /°C	W3 /%	T3 /°C	W4 /%	T4 /°C	Total Weight Loss / %	Remaining Weight/%
Blank	0	12.17	193.7	/	/	18.12	487.9	29.69	732.8	59.99	40.01
RAPD	0	12.03	196.3	2.22	431.1	15.88	495.2	29.50	730.3	59.64	40.36
RAP1	42.63	14.19	217.5	5.86	439.3	10.40	494.6	28.08	734.5	58.53	41.47
RAP2	77.46	17.16	219.5	6.06	444.6	9.91	507.9	26.46	734.5	59.58	40.42
RAP3	93.21	18.98	224.2	6.50	449.9	8.39	514.5	25.63	745.5	59.51	40.49

[*] W2/% represents the polysaccharide in the adsorbed crystals.

## Data Availability

All the data supporting the results were shown in the paper and can be applicable from the corresponding author.

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
