# Peer review of "Carboxymethylated Rhizoma alismatis Polysaccharides Regulate Calcium Oxalate Crystals Growth and Reduce the Regulated Crystals’ Cytotoxicity"

_biomolecules, 2023, doi:10.3390/biom13071044_

Round 1

Reviewer 1 Report

In this contribution Cheng and et. al. studied the effects of polysaccharides (RAP0) extracted from traditional anti-stone Chinese medicine Rhizoma alismatis and their carboxymethylated derivatives. The following manuscript is well written and the ideas are clear, however, it needs some corrections before being published:

On page 11, the 13C-NMR characterization discusses the carbon atom peaks and their number, however there is no figure that helps to better understand which carbon atom they are discussing. I suggest placing a structure with the atoms that are being numbered.

There is a format error that is repeated in all the text. It is related to the diffraction planes. This error begins on page 13 line 438 and continues on line 450. Page 15 line 460 and 461. Page 27 lines 721, 724, 725, 726 and 729. And in table 4. Finally, correct the formula every time potassium citrate appears.

The authors should give an explanation that the candidate compounds can be used as drugs since they present cytotoxicity when the size of the crystals changes. Is its potential viable, beyond the results obtained?

Author Response

Please see the  answer to Reviewer 1's question on the attachment.

Reviewer 2 Report

Dear Reviewers,

The manuscript written by Cheng X.Y., and Ouyang J.M., is well written and informative for the readers and could be a new addition in the sciences. The manuscript is suitable for publication but there are some minor errors/corrections need be addressed before publication.

All the corrections were mention in the pdf file with track signs.

Author Response

Please see the  answer to Reviewer 2's question on the attachment

Reviewer 3 Report

The article titled is devoted to the synthesis of new monosaccharide derivatives obtained from natural sources and their effect on the growth and toxicity of crystals in the kidneys. The language is quite good and the results will be of interest to the readers of the Journal. I believe that the article can be published after some edits.

1)      Is it necessary to duplicate the conclusions in the abstract of the article?

2)      Page 11. “Among these two peaks, the new peak appearing near d 80 ppm indicated that the carboxymethyl substitution might also occur at C-2 or C-4 position [45]. Meanwhile, the other new peak belonged to the carbonyl group introduced by carboxymethylation”. Carbonyl carbon should appear at 170 ppm, is there any signals in this area?

3)      Some more papers can be cited: [Oxidative Medicine and Cellular Longevity, 2020 , art 6982948, 10.1155/2020/6982948] [ACS Biomaterials Science & Engineering 2021 7 (7), 3409-3422 DOI: 10.1021/acsbiomaterials.1c00176] [Biomaterials Advances Volume 137, June 2022, 212854, 10.1016/j.bioadv.2022.212854] [Foods 2023, 12, 1031. 10.3390/foods12051031 ]

4)      Some technical notes:

1)      Page 1. “The -COOH contents of RAP0, RAP1, RAP2 and RAP3 were 7.79%, 10.84% and 15.33%, respectively”. But rap0 contain no COOH groups.

2)      In many techniques, the present simple is used. s. For example "Add sterile water to dissolve, filter into Thermo ICS-5000". Usually in the experimental part past simple and the passive voise used to be like: "sterile water was added to dissolve, and residue was filtered on Thermo ICS-5000"

3)      On the page 6: “, and distilled added to 24 mL” should be replaced with “, and distilled water added to 24 mL”

4)      Page 27 “Inhibit crystal aggregation” should be replaced with “Inhibition of crystal aggregation”

Author Response

Please see the  answer to Reviewer 3's question on the attachment

Reviewer 4 Report

Comments

Manuscript can be an interested to the wide range of scientific community. There are minor issues where the author need to concentrate prior to its final acceptance.

Title: both the titles are confusing and incomplete. Change it

Correspondence: authors.: Replace “authors” by “author”

Line 46-53:  Author must give citations for the statements

Line 206-217: Wrong. Check the writing format

Line 221-226: Remove A2, A1, and AO from the text.

Define A2, A1, and AO after equation.

“Each RAPs solution (3 mL) (concentrations of 1, 2, 4, 8, and 10 mg/mL, respectively) 221 and DPPH solution (1 mL, 0.4 mM, solvent: absolute ethanol) were mixed in a test tube. 222 The final concentration of DPPH was 0.1 mM; the reaction was performed at 25 °C for 30 223 min in the dark, and the absorbance was measured at 517 nm as A2. The detection absorb- 224 ance of 3 mL polysaccharide sample solution and 1 mL absolute ethanol mixture was A1; 225 the detection absorbance of 3 mL distilled water and 1 mL DPPH solution mixture was 226 A0. Positive control: Vc”.

Vc stands for? Define it

Line 204, 230, 238, 267: remove the citations from subheading. Include in the text.

, . Antioxidative activity of RAPs [25]

 Crystal growth and detection of soluble Ca2+ ions in the supernatant [25]

X-ray diffraction (XRD) characterization of CaOx crystals [25]

Preparation and characterization of FITC-RAP0 [35]

Which instruments you used to measures the values. Mention them in detail address.

Line 151: Give citation for “Sevage method.”

Explain the following methods in detail

“6). FT-IR characterization of RAPs

 Completely dried RAPs (2 mg) and KBr (200 mg) were ground and pressed together, and scanned in the wavenumber range of 4000-400 cm-1 with a resolution of 4 cm-1 . 196

7)1. H NMR and 13C NMR characterization of RAPs

 Completely dried RAPs (45 mg) was added to a nuclear magnetic resonance tube  filled with 0.6 mL of deuterated water (D2O), and after completely dissolved, nuclear mag-  netic resonance detection was performed at room temperature and pressure.

8). Zeta potential detection of RAPs

2 mg of RAPs was dissolved in 10 mL of ultrapure water to prepare a 0.20 mg/mL polysaccharide solution, and then the zeta potential was detected on the machine.”

3). FT-IR characterization of CaOx crystals

The experiment was the same as section2.2.6. Replaced RAPs with CaOx crystals.

4). Zeta potential detection of CaOx crystals CaOx crystals (5 mg) were added to 20.0 mL of pure water, then the zeta potential was measured.

5). Thermo gravimetric analysis (TGA) of CaOx crystals

Discussions

Avoid including figures and tables in the discussions section

Discussion can be improved by including more recent research work in the similar field.

Include the mechanism involve in the process.

Author Response

Please see the  answer to Reviewer 4's question on the attachment

Round 2

Reviewer 4 Report

The manuscript can be accepted in its present for.  The author has revised the manuscript and I am satisfied with the revision.